# Integrative phylogenetic analysis of the genus *Episoriculus* (Mammalia: Eulipotyphla: Soricidae)

**Yingxun Liu[1,2], Xuming Wang[1], Tao Wan[3], Rui Liao[1], Shunde Chen[3], Shaoying Liu[1]\*, Bisong Yue[2]\***

1 Sichuan Academy of Forestry, Chengdu, Sichuan, PR China, 2 College of Life Sciences, Sichuan University, Chengdu, Sichuan, PR China, 3 College of Life Sciences, Sichuan Normal University, Chengdu, Sichuan, PR China

\* shaoyliu@163.com (SL); bsyue@scu.edu.cn (BY)

**Data Availability Statement:** The datasets generated and analyzed during the current study contain potentially sensitive information and are

## Abstract

Shrews in the genus *Episoriculus* are among the least-known mammals in China, where representatives occur mainly in the Himalayan and Hengduan mountains. We sequence one mitochondrial and three nuclear genes from 77 individuals referable to this genus, collect morphometric data for five shape and 11 skull measurements from 56 specimens, and use museum collections and GenBank sequences to analyze phylogenetic relationships between this and related genera in an integrated molecular and morphometric approach. Whereas historically anywhere from two to eight species have been recognized in this genus, we conclude that six (*Episoriculus baileyi*, *E. caudatus*, *E. leucops*, *E. macrurus*, *E. sacratus*, *E. soluensis*) are valid. We dissent from recent systematic reviews of this genus and regard *E. sacratus* to be a valid taxon, *E. umbrinus* to be a subspecies of *E. caudatus*, and transfer *E. fumidus* to *Pseudosoriculus*. Our record of *E. soluensis* is the first for China, and expands the previously recognized distribution of this taxon from Nepal and NE India into the adjacent Yadong and Nyalam counties. One further undescribed *Episoriculus* taxon may exist in Xizang.

## Introduction

The genus *Episoriculus*, originally established as a subgenus of *Soriculus*, occurs throughout southwest China, India, Nepal, and Vietnam [1–9]. This subgenus was assigned full generic status by Repenning [10] on grounds of significant differences in tooth morphology from other species of *Chodsigoa* and *Soriculus*—a taxonomy followed by Jameson and Jones [11], Hutterer [12], Wilson and Reeder [13], and Wilson and Mittermeier [3].

The number of valid species of *Episoriculus* has been the subject of debate, with 2–8 species recognized (Table 1). Allen [14] described *Soriculus macrurus*, *S. caudatus sacratus*, and *S. caudatus umbrinus*. Ellerman and Morrison-Scott [1] proposed *Episoriculus* as a subgenus of *Soriculus*, and included *S. leucops* and *S. caudatus* (with subspecies *S. c. caudatus*, *S. c. baileyi*, *S. c. fumidus*, *S. c. sacratus*, and *S. c. umbrinus*). Honacki *et al.* [6] proposed that *Episoriculus*

not publicly available due to the potential for loss of privacy as required by the research ethics committee. but are available from the corresponding author and research ethics committee (irb@hsc.utah.edu) on reasonable request.

**Funding:** This research was funded by the National Natural Science Foundation of China (32370496, 31970399). Prof. Shaoying Liu, as the funder and corresponding author, participated in study design, data collection and analysis, decision to publish, or preparation of the manuscript.

**Competing interests:** The authors have declared that no competing interests exist.

included four species, and considered *S. baileyi* and *S. fumidus* to be valid taxa. Hoffmann [7] similarly recognized four species, although these were not entirely consistent with those of Honacki *et al.* [6], for *S. baileyi* was relegated to a subspecies of *S. (E.) leucops*, and *S. (E.) macrurus* was placed in this genus. Corbet and Hill [8], and Wilson and Reeder [9, 13] followed this arrangement. Motokawa and Lin [15] elevated *S. baileyi* to full species based on morphology. Based on the karyotypes and differences in skull morphology, Motokawa *et al.* [16] considered that *Episoriculus caudatus* should be divided into the larger *E. caudatus* and smaller *E. sacratus* (with subspecies *E. s. soluensis* from Nepal and Sikkim, *E. s. umbrinus* from Assam, Myanmar, and Yunnan, China, and *E. s. sacratus* from Sichuan, China). He *et al.* [17] noted that *Pseudosoriculus fumidus* did not belong to *Episoriculus*. Based on *CYTB* gene sequences, Abramov *et al.* [2] promoted *E. soluensis* to full species, assigned *E. fumidus* to a new genus *Pseudosoriculus*, and arranged seven species (*E. baileyi*, *E. caudatus*, *E. leucops*, *E. macrurus*, *E. sacratus*, *E. soluensis*, and *E. umbrinus*) in *Episoriculus*. Wilson and Mittermeier [3] recognized eight species, including *P. fumidus*.

Throughout these various classifications the taxonomic status of *E. caudatus*, *E. leucops*, and *E. macrurus* has been relatively stable, but the taxonomy of *P. fumidus*, *E. sacratus*, E. umbrinus, *E. baileyi*, and *E. soluensis* has not. We report new molecular data and morphological comparisons in an integrated phylogenetic and morphological analysis to clarify the taxonomic status of species in the genus *Episoriculus*.

## Materials and methods

### Field methodology

In the study, traplines were installed to capture shrews. We use red plastic buckets as traps. Each of these 7L buckets was 22 cm high and had an upper and lower diameter of 26 cm and 19 cm, respectively. We typically position these barrels in the more humid forest, primarily around the roots of massive, or falling trees, 3-5m apart. The buckets were often put in the afternoon of the first day and checked the following morning to see whether any shrews have fallen into the trap. The time between placement and inspection is assured to be more than 12 hours.

After euthanasia with eugenol, following the ASM guidelines [18], tissue samples were taken from the thigh muscle and stored in absolute ethyl alcohol at ambient temperature. Voucher specimens fixed in 10% formalin before transferal to 95% ethanol for long term preservation and deposition of vouchers at the Zoological Museum of Sichuan Academy of Forestry (SAF), Chengdu, Sichuan, China. All specimens were collected in accordance with regulations in China for implementation of the protection of terrestrial wild animals (State Council Decree [1992] No. 13). Collecting protocols and the research project were approved by the Ethics Committee of Sichuan Academy of Forestry (2024–001).

### Sampling and sequencing

All specimens were identified morphologically following the original literature species established [19–23], and some subsequent studies on this group, such as Wilson and Mittermeier [3], Hoffmann [7], and Smith and Xie [24] in which, some specimens which were be identified as *E. sacratus* and *E. soluensis*, were collected adjacent of the type locality (Table 2). For verifying our species accuracies, some sequences collected form type locality or adjacent of the type locality, were downloaded [2, 17, 25–32] (Table 3). Of recognized species, we had no specimens or sequences of *E. baileyi*. Through preliminary molecular and morphological identification, 77 specimens collected from China, 31 were attributed to *E. macrurus*, 18 to *E. caudatus*, 10 to *E. leucops*, 5 to *E. umbrinus*, 4 of each to *E. sacratus* and *E. soluensis*, and 2 to *E.* sp. (Table 1 and Fig 1)

**Table 1. Major classification systems of the genus *Episoriculus*.** A name of the species in parentheses indicates that the taxon is a subspecies of the previous species.

| Allen [14] | Ellerman and Morrison-Scott [1] | Honacki *et al.* [6] | Hoffmann [7] | Wilson and Reeder [9] | Wilson and Reeder [13] | Abramov *et al.* [2] | Wilson and Mittermeier [3] |
|---|---|---|---|---|---|---|---|
| S. caudatus | S. caudatus | S. baileyi | S. caudatus | S. caudatus | E. caudatus | E. caudatus | E. caudatus |
| (umbrinus) | (umbrinus) | S. caudatus | S. leucops | S. baileyi | (sacratus) | E. umbrinus | E. umbrinus |
| (sacratus) S. leucops | (baileyi) | S. fumidus | (baileyi) | S. leucops | (umbrinus) | E. sacratus | E. sacratus |
|  | (sacratus) | S. leucops | S. macrurus | S. macrurus | E. fumidus | E. leucops | E. leucops |
|  | (fumidus) |  | S. fumidus | S. fumidus | E. macrurus | E. macrurus | E. macrurus |
|  | S. leucops |  |  |  | E. leucops | E. soluensis | E. soluensis |
|  |  |  |  |  | (baileyi) | E. baileyi | E. fumidus |
|  |  |  |  |  |  |  | E. baileyi |

Following He *et al*. [17] and Chen *et al*. [32, 33], we amplified the complete mitochondrial cytochrome b (*CYTB*) and three partial nuclear genes (apolipoprotein B (*APOB*), recombination-activating gene 2 (*RAG2*), and breast cancer 1 (*BRCA1*)). Primer sets are detailed in Table 4 [17, 30, 34, 35]. PCR amplifications were carried out in a 25 µl reaction volume mixture containing 12.5 µl of 2×Taq Master Mix (Vazyme, Nanjing, China), 1 µl of each primer, 1 µl of genomic DNA, and 9.5 µl of double-distilled water. PCR conditions for *CYTB* amplifications consisted of an initial denaturing step at 94˚C for 5 min followed by 38 cycles of denaturation at 94˚C for 45 s, annealing at 49˚C for 45 s, an extension at 72˚C for 90 s, and a final extension step at 72˚C for 12 min. PCR conditions for nuclear genes were basically the same as those for *CYTB*, with a few modifications (annealing temperatures for each nuclear gene were *APOB* (49˚C), *BRCA1* (51˚C), and *RAG2* (52˚C)). PCR products were checked on a 1.0% agarose gel and purified by ethanol precipitation. Purified PCR products were directly sequenced using the BigDye Terminator Cycle Kit v 3.1 (Applied Biosystems, Foster City, CA, USA) and an ABI 310 Analyzer (Applied Biosystems).

To test phylogenetic relationships within the genus *Episoriculus*, sequences of these four genes from other Nectogalini species generated in previous studies [2, 17, 25–32] were downloaded from GenBank (Table 2).

## Sequence analyses

All *CYTB* sequences were aligned and examined. Screening for heterozygous nuclear gene fragments was performed in Mega 5 [36]. For analysis, we concatenated the three nuclear genes following He *et al*. [17]. Using all sequence data, phylogenetic analyses were conducted on: 1) *CYTB* data, 2) concatenated sequences for the three nuclear genes, and 3) each nuclear gene. Modeltest v 3.7 [37] was used to select the best-fitting evolutionary model, based on the Akaike Information Criterion in Table 3. MrBayes v 3.1.2 [38] was used for Bayesian analysis. *Crocidura fuliginosa* was selected as the outgroup. Each run was carried out with four Monte Carlo Markov chains (MCMCs), and 10,000,000 generations for single gene datasets and 30,000,000 generations for concatenated gene datasets. All runs were sampled every 10,000 generations. Convergences of runs were accepted when the average standard deviation of split frequencies was $< 0.01$. Ultrafast bootstrap values (UFBoot) of $\geq 95$ and posterior probabilities (PP) of $\geq 0.95$ were considered strong support [39].

**Table 2. Samples and sequences of *Episoriculus* used for molecular analyses.**

| Species | Species ID | Sequence No. | Locality | Relative location | longitude | Latitude | Altitude (m) | Genbank accession No. | | | |
|---|---|---|---|---|---|---|---|---|---|---|---|
| | | | | | | | | *CYTB* | *APOB* | *BRCA1* | *RAG2* |
| *Episoriculus sacratus* | SAF15216 | LS15073 | Lushan, Sichuan | | 103.04592 | 30.66332 | 2018 | MK962202 | / | MN032211 | MN032279 |
| *Episoriculus sacratus* | SAF06888 | ELSTNPB01007 | Erlangshan, Sichuan | adjacent to type locality | 102.29916 | 29.88547 | 2450 | MK962204 | MN032147 | MN032213 | MN032281 |
| *Episoriculus sacratus* | SAF06902 | ELSTNPB02004 | Erlangshan, Sichuan | adjacent to type locality | 102.29916 | 29.88547 | 2450 | MK962207 | MN032149 | MN032214 | MN032283 |
| *Episoriculus sacratus* | SAF11594 | BLG012 | Balanggou, Sichuan | adjacent to type locality | 101.98069 | 30.41007 | 2460 | MK962218 | / | MN032223 | / |
| *Episoriculus sacratus* | SAF06934 | ELS260B02003 | Erlangshan, Sichuan | adjacent to type locality | 102.31112 | 29.87314 | 2230 | MK962203 | / | MN032212 | MN032280 |
| *Episoriculus sacratus* | SAF06920 | ELS260A02003 | Erlangshan, Sichuan | adjacent to type locality | 102.30982 | 29.87294 | 2220 | MK962217 | / | MN032222 | MN032292 |
| *Episoriculus umbrinus* | SAF13199 | NJ13033 | Lushui, Yunnan | | 98.70969 | 25.98614 | 2719 | MK962237 | MN032174 | MN032241 | MN032310 |
| *Episoriculus umbrinus* | SAF13257 | NJ13091 | Gongshan, Yunnan | | 98.44691 | 27.77396 | 3320 | MK962198 | / | MN032207 | / |
| *Episoriculus umbrinus* | SAF180044 | YN18021 | Mountain Gaoligong, Yunnan | | 98.7111 | 25.97795 | 2570 | MK962223 | MN032163 | MN032228 | MN032297 |
| *Episoriculus caudatus* | SAF07474 | XZRAP05005 | Nyingchi, Xizang | | 94.962862 | 30.00965 | 2040 | MK962213 | MN032155 | MN032218 | MN032289 |
| *Episoriculus caudatus* | SAF07492 | XZRAP07004 | Nyingchi, Xizang | | 94.9631 | 30.00942 | 2020 | MK962212 | MN032154 | MN032217 | MN032288 |
| *Episoriculus caudatus* | SAF07568 | XZXCY04003 | Xiachayu, Xizang | | 97.01193 | 28.51268 | 1640 | MK962224 | / | MN032229 | MN032298 |
| *Episoriculus caudatus* | SAF07483 | XZRAP06001 | Nyingchi, Xizang | | 94.95855 | 30.00766 | 2050 | MK962214 | MN032156 | MN032219 | MN032290 |
| *Episoriculus caudatus* | SAF07566 | XZXCY04001 | Xiachayu, Xizang | | 97.01193 | 28.51268 | 1640 | MK962206 | / | / | / |
| *Episoriculus caudatus* | SAF07502 | XZRAP08004 | Nyingchi, Xizang | | 94.95878 | 30.01025 | 2040 | MK962219 | MN032159 | MN032224 | MN032293 |
| *Episoriculus caudatus* | SAF07501 | XZRAP08003 | Nyingchi, Xizang | | 94.95878 | 30.01025 | 2040 | MK962222 | MN032162 | MN032227 | MN032296 |
| *Episoriculus caudatus* | SAF07472 | XZRAP05003 | Nyingchi, Xizang | | 94.962862 | 30.00965 | 2040 | MK962221 | MN032161 | MN032226 | MN032295 |
| *Episoriculus caudatus* | SAF07490 | XZRAP07002 | Nyingchi, Xizang | | 94.9631 | 30.00942 | 2020 | MK962215 | MN032157 | MN032220 | / |
| *Episoriculus caudatus* | SAF07535 | XZLZT02001 | Nyingchi, Xizang | | 95.00633 | 30.02289 | 2020 | MK962211 | MN032153 | / | MN032287 |
| *Episoriculus caudatus* | SAF07470 | XZRAP05001 | Nyingchi, Xizang | | 94.962862 | 30.00965 | 2040 | MK962216 | MN032158 | MN032221 | MN032291 |
| *Episoriculus* sp. | SAF11247 | MT11162 | Motuo, Xizang | | 95.4814 | 29.43014 | 2832 | MK962200 | MN032145 | MN032209 | MN032277 |
| *Episoriculus* sp. | SAF11283 | MT11198 | Motuo, Xizang | | 95.0095 | 29.40422 | 3118 | MK962199 | MN032144 | MN032208 | MN032276 |
| *Episoriculus leucops* | SAF180058 | YN18035 | Mountain Gaoligong, Yunnan | | 98.7145 | 25.95919 | 2250 | MK962232 | MN032169 | MN032237 | MN032306 |

(*Continued*)

**Table 2.** (Continued)

| Species | Species ID | Sequence No. | Locality | Relative location | longitude | Latitude | Altitude (m) | Genbank accession No. | | | |
|---|---|---|---|---|---|---|---|---|---|---|---|
| | | | | | | | | *CYTB* | *APOB* | *BRCA1* | *RAG2* |
| *Episoriculus leucops* | SAF180035 | YN18012 | Mountain Gaoligong, Yunnan | | 98.7111 | 25.97795 | 2570 | MK962231 | MN032168 | | MN032305 |
| *Episoriculus leucops* | SAF13190 | NJ13024 | Lushui, Yunnan | | 98.68388 | 25.97259 | 3150 | MK962234 | MN032171 | MN032239 | MN032308 |
| *Episoriculus leucops* | SAF180036 | YN18013 | Mountain Gaoligong, Yunnan | | 98.7111 | 25.97795 | 2570 | MK962227 | MN032165 | MN032232 | MN032301 |
| *Episoriculus leucops* | SAF180033 | YN18010 | Mountain Gaoligong, Yunnan | | 98.7111 | 25.97795 | 2570 | MK962229 | MN032167 | MN032234 | MN032303 |
| *Episoriculus leucops* | SAF180034 | YN18011 | Mountain Gaoligong, Yunnan | | 98.7111 | 25.97795 | 2570 | MK962233 | MN032170 | MN032238 | MN032307 |
| *Episoriculus leucops* | SAF180045 | YN18022 | Mountain Gaoligong, Yunnan | | 98.7111 | 25.97795 | 2570 | MK962226 | / | MN032231 | MN032300 |
| *Episoriculus leucops* | SAF180040 | YN18017 | Mountain Gaoligong, Yunnan | | 98.7111 | 25.97795 | 2570 | MK962228 | MN032166 | MN032233 | MN032302 |
| *Episoriculus leucops* | SAF180037 | YN18014 | Mountain Gaoligong, Yunnan | | 98.7111 | 25.97795 | 2570 | MK962230 | / | MN032235 | MN032304 |
| *Episoriculus leucops* | SAF11202 | MT11017 | Motuo, Xizang | | 95.126722 | 29.3665 | 2100 | MK962235 | MN032172 | MN032240 | MN032309 |
| *Episoriculus macrurus* | SAF180039 | YN18016 | Mountain Gaoligong, Yunnan | | 98.7111 | 25.97795 | 2570 | MK962166 | / | MN032180 | MN032247 |
| *Episoriculus macrurus* | SAF13198 | NJ13032 | Lushui, Yunnan | | 98.70969 | 25.98614 | 2719 | MK962170 | MN032126 | MN032184 | MN032251 |
| *Episoriculus macrurus* | SAF13256 | NJ13090 | Gongshan, Yunnan | | 98.44691 | 27.77396 | 3320 | MK962165 | / | MN032179 | MN032246 |
| *Episoriculus macrurus* | SAF13288 | NJ13122 | Gongshan, Yunnan | | 98.50351 | 27.79702 | 3050 | MK962167 | / | MN032181 | MN032248 |
| *Episoriculus macrurus* | SAF13197 | NJ13031 | Lushui, Yunnan | | 98.70969 | 25.98614 | 2719 | MK962163 | / | MN032177 | / |
| *Episoriculus macrurus* | SAF07587 | CY17 | Chayu, Xizang | | 97.01826 | 28.77404 | 2900 | MK962236 | MN032173 | / | / |
| *Episoriculus macrurus* | SAF09629 | JJSA506 | Jinjiashan, Sichuan | | 102.75222 | 30.84041 | 2620 | MK962195 | / | MN032205 | MN032274 |
| *Episoriculus macrurus* | SAF181173 | WL18246 | Wanglang, Sichuan | | 104.1342 | 32.94794 | 2570 | MK962179 | MN032134 | / | MN032260 |
| *Episoriculus macrurus* | SAF06900 | ELSTNPB02002 | Erlangshan, Sichuan | | 102.29916 | 29.88547 | 2450 | MK962175 | MN032131 | MN032188 | MN032256 |
| *Episoriculus macrurus* | SAF06829 | ELSMYPA02004 | Erlangshan, Sichuan | | 102.28433 | 29.86774 | 2780 | MK962162 | / | MN032176 | MN032244 |
| *Episoriculus macrurus* | SAF11599 | BLG031 | Balanggou, Sichuan | | 101.91481 | 30.40052 | 2820 | MK962187 | MN032139 | MN032197 | / |
| *Episoriculus macrurus* | SAF06932 | ELS260B02001 | Erlangshan, Sichuan | | 102.31112 | 29.87314 | 2230 | MK962173 | / | MN032187 | MN032254 |
| *Episoriculus macrurus* | SAF06874 | ELSTNPA02002 | Erlangshan, Sichuan | | 102.29953 | 29.88633 | 2430 | MK962176 | / | MN032189 | MN032257 |

(*Continued*)

**Table 2.** (Continued)

| Species | Species ID | Sequence No. | Locality | Relative location | longitude | Latitude | Altitude (m) | Genbank accession No. | | | |
|---|---|---|---|---|---|---|---|---|---|---|---|
| | | | | | | | | *CYTB* | *APOB* | *BRCA1* | *RAG2* |
| *Episoriculus macrurus* | SAF181086 | WL18159 | Wanglang, Sichuan | | 104.1437 | 32.928634 | 2500 | MK962180 | MN032135 | / | MN032261 |
| *Episoriculus macrurus* | SAF181215 | WL18288 | Wanglang, Sichuan | | 117.1342 | 45.94794 | 2583 | MK962183 | / | MN032193 | MN032263 |
| *Episoriculus macrurus* | SAF181127 | WL18200 | Wanglang, Sichuan | | 104.1459 | 32.927797 | 2500 | MK962189 | MN032140 | MN032199 | MN032268 |
| *Episoriculus macrurus* | SAF181126 | WL18199 | Wanglang, Sichuan | | 104.1459 | 32.927797 | 2500 | MK962188 | / | MN032198 | MN032267 |
| *Episoriculus macrurus* | SAF181065 | WL18138 | Wanglang, Sichuan | | 104.1437 | 32.928634 | 2500 | MK962185 | / | MN032195 | MN032265 |
| *Episoriculus macrurus* | SAF181130 | WL18203 | Wanglang, Sichuan | | 104.1459 | 32.927797 | 2500 | MK962184 | MK962184 | MN032194 | MN032264 |
| *Episoriculus macrurus* | SAF181129 | WL18202 | Wanglang, Sichuan | | 104.1459 | 32.927797 | 2500 | MK962190 | MN032141 | MN032200 | MN032269 |
| *Episoriculus macrurus* | SAF09655 | JJSA532 | Balanggou, Sichuan | | 102.75311 | 30.84053 | 2620 | MK962193 | MN032143 | MN032203 | MN032272 |
| *Episoriculus macrurus* | SAF181137 | WL18137 | Wanglang, Sichuan | | 104.1459 | 32.927797 | 2500 | MK962182 | MN032136 | MN032192 | / |
| *Episoriculus macrurus* | SAF06725 | ELSLCA03001 | Erlangshan, Sichuan | | 102.26703 | 29.85678 | 2800 | MK962172 | / | MN032186 | MN032253 |
| *Episoriculus macrurus* | SAF181394 | WL18467 | Wanglang, Sichuan | | 103.9963 | 32.96181 | 3000 | MK962174 | MN032130 | / | MN032255 |
| *Episoriculus macrurus* | SAF181128 | WL18201 | Wanglang, Sichuan | | 104.1459 | 32.927797 | 2500 | MK962181 | / | MN032191 | MN032262 |
| *Episoriculus macrurus* | SAF15165 | LS15023 | Lushan, Sichuan | | 103.01808 | 30.68135 | 2424 | MK962194 | / | MN032204 | MN032273 |
| *Episoriculus macrurus* | SAF06730 | ELSDGA01003 | Erlangshan, Sichuan | | 102.2584 | 29.85416 | 2700 | MK962196 | / | MN032206 | MN032275 |
| *Episoriculus macrurus* | SAF06767 | ELSBTDBS01013 | Erlangshan, Sichuan | | 102.26283 | 29.85202 | 2500 | MK962161 | / | MN032175 | MN032243 |
| *Episoriculus macrurus* | JJSA137 | JJSA137 | Jiajinshan, Sichuan | | | | | MK962192 | MN032142 | MN032202 | MN032271 |
| *Episoriculus macrurus* | SAF181202 | WL18275 | Wanglang, Sichuan | | 104.1342 | 32.94794 | 2570 | MK962178 | MN032133 | / | MN032259 |
| *Episoriculus macrurus* | SAF09303 | JJSA443 | Jiajinshan, Sichuan | | 10268985 | 30.84276 | 3440 | MK962186 | MN032138 | MN032196 | MN032266 |
| *Episoriculus macrurus* | HLG01017 | HLG01017 | Hailuogou, Sichuan | | | | | MK962191 | / | MN032201 | MN032270 |
| *Episoriculus macrurus* | SAF06177 | MGDW0502 | Meigu, Sichuan | | 103.27607 | 28.7063 | 2900 | MK962177 | MN032132 | MN032190 | MN032258 |
| *Episoriculus soluensis* | SAF13503 | XZ13041 | Nyalam, Xizang | adjacent to type locality | 85.99854 | 28.0815 | 3200 | MK962201 | MN032146 | MN032210 | MN032278 |
| *Episoriculus soluensis* | SAF14536 | XZ14002 | Yadong, Xizang | | 88.994744 | 27.540042 | 4304 | MK962157 | / | / | / |
| *Episoriculus soluensis* | SAF14537 | XZ14003 | Yadong, Xizang | | 88.994744 | 27.540042 | 4304 | MK962158 | / | / | / |
| *Episoriculus soluensis* | SAF14538 | XZ14004 | Yadong, Xizang | | 88.994744 | 27.540042 | 4304 | MK962159 | / | / | / |

**Table 3. GenBank accession numbers of download sequence from NCBI.**

| Species | Locality | Relative location | CYTB | APOB | BRCA1 | RAG2 | Cited source |
|---------|----------|-------------------|------|------|-------|------|--------------|
| *Anourosorex squamipes* | Baoxing, Sichuan, China | type locality | KT032922 | / | / | / | He *et al.* [25] |
| *Anourosorex squamipes* | Baoxing, Sichuan, China | type locality | KT032921 | / | / | / | He *et al.* [25] |
| *Anourosorex squamipes* | Baoxing, Sichuan, China | type locality | KT032920 | / | / | / | He *et al.* [25] |
| *Anourosorex squamipes* | Baoxing, Sichuan, China | type locality | KT032919 | / | / | / | He *et al.* [25] |
| *Blarinella griselda* | Yunnan, China | | GU981259 | GU981109 | GU981184 | / | He *et al.* [17] |
| *Blarinella griselda* | Yunnan, China | | GU981258 | GU981108 | GU981183 | GU981441 | He *et al.* [17] |
| *Blarinella quadraticauda* | Baoxing, Sichuan, China | type locality | JF719718 | / | / | / | Chen *et al.* [26] |
| *Blarinella quadraticauda* | Baoxing, Sichuan, China | type locality | JF719719 | / | / | / | Chen *et al.* [26] |
| *Blarinella quadraticauda* | Baoxing, Sichuan, China | type locality | JF719720 | / | / | / | Chen *et al.* [26] |
| *Blarinella quadraticauda* | Baoxing, Sichuan, China | type locality | JF719721 | / | / | / | Chen *et al.* [26] |
| *Chimarrogale himalayica* | Yunnan, China | adjacent to type locality | GU981264 | GU981112 | GU981187 | GU981445 | He *et al.* [17] |
| *Chimarrogale himalayica* | Yunnan, China | adjacent to type locality | GU981263 | / | / | / | He *et al.* [17] |
| *Chodsigoa hypsibia* | Beichuan, Sichuan, China | adjacent to type locality | KX765527 | / | / | / | Chen *et al.* [27] |
| *Chodsigoa hypsibia* | Mengda, Qinghai, China | | KX765528 | / | / | / | Chen *et al.* [27] |
| *Chodsigoa hypsibia* | Mengda, Qinghai, China | | KX765529 | / | / | / | Chen *et al.* [27] |
| *Chodsigoa hypsibia* | Mengda, Qinghai, China | | KX765530 | / | / | / | Chen *et al.* [27] |
| *Chodsigoa hypsibia* | Mengda, Qinghai, China | | KX765533 | / | / | / | Chen *et al.* [27] |
| *Crocidura fuliginosa* | Yunnan, China | adjacent to type locality | GU981271 | GU981117 | GU981192 | GU981450 | He *et al.* [17] |
| *Crossogale hantu* | Selangor, Malaysia | | MN149424 | MN149427 | / | MN149430 | Abd Wahab *et al.* [28] |
| *Crossogale hantu* | Selangor, Malaysia | | MN149423 | MN149426 | / | MN149429 | Abd Wahab *et al.* [28] |
| *Crossogale hantu* | Selangor, Malaysia | | MN149422 | MN149428 | / | MN149428 | Abd Wahab *et al.* [28] |
| *Episoriculus caudatus* | Yunnan, China | | GU981272 | GU981118 | GU981193 | GU981451 | He *et al.* [17] |
| *Episoriculus caudatus* | Yunnan, China | | GU981273 | / | / | / | He *et al.* [17] |
| *Episoriculus caudatus* | Yunnan, China | | GU981274 | GU981119 | GU981194 | GU981452 | He *et al.* [17] |
| *Episoriculus caudatus* | Yunnan, China | | GU981275 | / | / | / | He *et al.* [17] |
| *Episoriculus caudatus* | Yunnan, China | | GU981276 | GU981120 | GU981195 | GU981453 | He *et al.* [17] |
| *Episoriculus caudatus* | Yunnan, China | | GU981277 | / | / | / | He *et al.* [17] |
| *Episoriculus caudatus* | Phulchauki, Nepal | | AB175114 | / | / | / | Ohdachi *et al.* [29] |
| *Episoriculus caudatus* | Kurumsan, Nepal | | AB175115 | / | / | / | Ohdachi *et al.* [29] |
| *Episoriculus leucops* | Yunnan, China | | GU981281 | GU981122 | GU981197 | GU981455 | He *et al.*[17] |
| *Episoriculus leucops* | Yunnan, China | | GU981282 | GU981123 | GU981198 | GU981456 | He *et al.* [17] |
| *Episoriculus leucops* | Yunnan, China | | GU981283 | / | / | / | He *et al.* [17] |
| *Episoriculus leucops* | Yunnan, China | | GU981284 | / | / | / | He *et al.* [17] |
| *Episoriculus leucops* | Syng Gomba, Nepal | | AB175111 | / | / | / | Ohdachi *et al.* [29] |
| *Episoriculus macrurus* | Yunnan, China | | GU981285 | GU981122 | GU981197 | GU981455 | He *et al.* [17] |
| *Episoriculus macrurus* | Yunnan, China | | GU981286 | GU981123 | GU981198 | GU981456 | He *et al.* [17] |
| *Episoriculus macrurus* | Yunnan, China | | GU981287 | / | / | / | He *et al.* [17] |
| *Episoriculus macrurus* | Yunnan, China | | GU981288 | / | / | / | He *et al.* [17] |
| *Episoriculus macrurus* | Yunnan, China | | GU981289 | GU981124 | GU981199 | GU981457 | He *et al.* [17] |
| *Episoriculus macrurus* | Yunnan, China | | GU981290 | GU981125 | GU981200 | GU981458 | He *et al.* [17] |
| *Episoriculus macrurus* | Pokhara, Nepal | | AB175109 | / | / | / | Ohdachi *et al.* [29] |
| *Episoriculus macrurus* | Syabru, Nepal | | AB175110 | / | / | / | Ohdachi *et al.* [29] |
| *Episoriculus soluensis* | Gosainkund, Nepal | adjacent to type locality | AB175112 | / | / | / | Ohdachi *et al.* [29] |
| *Episoriculus soluensis* | Gosainkund, Nepal | adjacent to type locality | AB175113 | / | / | / | Ohdachi *et al.* [29] |
| *Nectogale elegans* | Yunnan, China | | GU981294 | / | / | / | He *et al.*[17] |
| *Nectogale elegans* | Yunnan, China | | GU981293 | GU981129 | GU981204 | GU981462 | He *et al.* [17] |
| *Neomys fodiens* | Popova Sapka, North Macedonia | | / | DQ630162 | DQ630245 | / | Dubey *et al.*[30] |

(*Continued*)

**Table 3.** (Continued)

| Species | Locality | Relative location | *CYTB* | *APOB* | *BRCA1* | *RAG2* | Cited source |
|---|---|---|---|---|---|---|---|
| *Neomys fodiens* | Hochsauerlandkreis, Germany | adjacent to type locality | / | GU981130 | GU981205 | GU981463 | He *et al.*[17] |
| *Pantherina griselda* | Sichuan, China | | KY249527 | KY249532 | MN199109 | KY249544 | Bannikova *et al.*[31] |
| *Pseudosoriculus fumidus* | Taiwan, China | adjacent to type locality | / | DQ630193 | DQ630273 | / | Dubey *et al.* [30] |
| *Pseudosoriculus fumidus* | Taiwan, China | adjacent to type locality | GU981278 | GU981121 | GU981196 | GU981454 | He *et al.*[17] |
| *Pseudosoriculus fumidus* | Taiwan, China | adjacent to type locality | GU981279 | / | / | / | He *et al.*[17] |
| *Pseudosoriculus fumidus* | Taiwan, China | adjacent to type locality | GU981280 | / | / | / | He *et al.* [17] |
| *Pseudosoriculus fumidus* | Taiwan, China | adjacent to type locality | AB175107 | / | / | / | Ohdachi *et al.* [29] |
| *Pseudosoriculus fumidus* | Taiwan, China | adjacent to type locality | AB175108 | / | / | / | Ohdachi *et al.* [29] |
| *Sorex bedfordiae* | Pingwu, Sichuan, China | | OL585127 | OL585986 | OL585694 | OL586703 | Chen *et al.* [32] |
| *Sorex bedfordiae* | Pingwu, Sichuan, China | | OL585125 | OL585984 | OL585692 | OL586701 | Chen *et al.* [32] |
| *Sorex bedfordiae* | Pingwu, Sichuan, China | | OL585126 | OL585985 | OL585693 | OL586702 | Chen *et al.* [32] |
| *Sorex bedfordiae* | Pingwu, Sichuan, China | | OL585128 | OL585987 | OL585695 | OL586704 | Chen *et al.* [32] |
| *Soriculus nigrescens* | Yunnan, China | | GU981297 | / | / | / | He *et al.* [17] |
| *Soriculus nigrescens* | Yunnan, China | | GU981298 | GU981132 | GU981207 | GU981465 | He *et al.* [17] |

## Species tree and species delimitation

To appraise current taxonomic systems based on morphology, and to evaluate trees derived from phylogenetic analyses, we used species delimitation and species tree construction based on coalescence theory [40, 41]. The basis of abductive theory-based species definition is to use the multispecies coalescent model to identify evolutionarily independent lineages in genomic data. which is significantly distinct from defining species as stable differences between monophyletic groups or taxa [42]. Since the *BEAST model requires that every gene segment in each sample be complete, we didn't use all samples. We estimated the *BEAST coalescent species tree in Beast v 1.7.5 using partial nuclear and mitochondrial genes [43, 44]. Model settings were selected with reference to the optimal replacement model of each gene (Table 3). Each

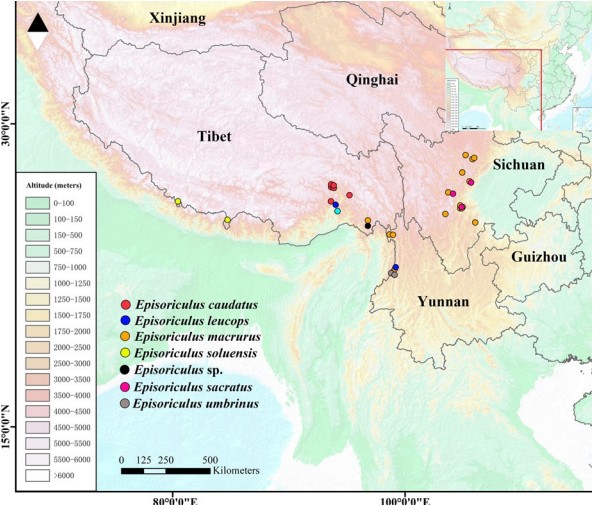

**Fig 1. Map of genus *Episoriculus*, showing localities sampled for this study.** Reprinted from Fan *et al.* (2022) [Doi: https://doi.org/10.1002/ece3.9404] under a CC BY license, with permission from Fan Ronghui, original copyright 2022. And edited and supplemented by Adobe Photoshop 2022 and Arcgis 10.8.

**Table 4. Gene symbol, primer sequences and the best model of evolution for each gene segments used in the study.**

| Genes | Primers sequences (5'→3') | The best model | Cited source |
|---|---|---|---|
| CYTB | GGACTTATGACATGAAAAATCATCGTTG | GTR+I+G | He *et al.* [17] |
| | GATTCCCCATTTCTGGTTTACAAGAC | | He *et al.* [17] |
| APOB | GCAATCATTTGACTTAAGTG | HKY+G | Dubey *et al.* [30] |
| | GAGCAACAATATCTGATTGG | | Dubey *et al.* [30] |
| BRCA1 | TGAGAACAGCACTTTATTACTCAC | HKY+I+G | Dubey *et al.* [34] |
| | ATTCATGTTCCATATTGCTTATACTG | | Dubey *et al.* [34] |
| RAG2 | AGCTGCAGYCARTACCAYAARATGTA | HKY+I+G | Murphy *et al.* [35] |
| | AACTCAGCTGCATTKCCAATRTCACA | | Murphy *et al.* [35] |

analysis was run for $100 \times 10^6$ generations and sampled every 10,000 generations. Posterior distribution and effective sample size of each parameter were calculated using Tracer v 1.6. *BEAST analyses were repeated four times to ensure convergence on the same posterior distribution.

"Splits" (Species Limits by Threshold Statistics) v 1.0 14 was used for species delimitation in the context of R statistics. The program defines species using the generalized mixed Yule-coalescent model (GMYC) [45, 46]. Analysis requires a gene tree that has been corrected by a molecular clock as a reference; we use the bifurcated time tree constructed with Beast v 1.7.5 as basic data for this analysis.

BPP v 2.2 was used to species delimitation [47, 48]. Analyses of species boundaries were limited to *E. caudatus*, *E. sacratus*, and *E. umbrinus*; analyses used the combined nuclear gene data set. Since this program also requires a prior guide tree, and the topological structure of the guide tree will impact the result of species delimitation, we used the aforementioned species tree built by BEAST v 1.7.5 as the guide tree for BPP analysis [49]. For BPP, we set the Gamma prior distribution of the population size parameter (θs) to G (6, 6,000), and the initial differentiation time parameter (τ0) of the species tree to G (4, 1,000). Then 12 parameter combinations were generated using algorithms 0 and 1, and combining the values of Locusrate = 1 or Heredity = 1. The above 12 operations were performed on the two data groups; each operation was set to 1,000,000 generations of reverse-jump MCMC, and samples were taken every 10 generations; the first 10,000 generations were discarded [50].

Automatic barcode gap discovery (ABGD) software was used to divide samples based on genetic distance; samples within the same group were identified as one species [51]. *CYTB* sequences of the sample online submission to ABGD website (http://wwwabi.snv.Jussleu.fr/publicabgd/abgdweb.hml), the prior intraspecific divergence (P) ranged 0.001–0.1, and the minimum relative gap width (X) was 0.5.

We also used the Kimura 2-parameter (K2p) distance with 10,000 bootstrap replicates to summarize sequence divergences based on *CYTB* in MEGA5 [36].

## Morphometrics

Because we have found no evidence for sexual dimorphism in shrews in related taxa, we do not consider sex when selecting specimens and skulls [52]. All samples used for analysis were adults with intact skulls. Specimens of *Episoriculus* used for study have been deposited in the Sichuan Academy of Forestry, and Kunming Institute of Zoology. A total of 56 complete skulls of adult intact specimens were assigned to *E. macrurus* (17), *E. caudatus* (11), *E. umbrinus* (8), *E. leucops* (7), *E. sacratus* (6), and *E. soluensis* (3). Details of localities and museums are listed in S1 Table.

Measurements including head and body length (HBL), tail length (TL), hind foot length (HFL), and ear length (EL) were recorded from specimen labels or field notes. We measured the skulls of these specimens with a digital Vernier caliper (accuracy 0.01 mm). Eleven cranio-mandibular variables were taken: profile length (PL), cranial height (CH), cranial breadth (CB), interorbital breadth (IBO), palatoincisive length (PIL), postpalatal length (PPL), maxillary breadth (MB), upper toothrow length (UTR), maximum width across the upper second molars ($M^2$–$M^2$), mandibular length (ML), and lower toothrow length (LTR). Measurements dates were used for principal component analysis (PCA) in SPSS 22.0 (SPSS Inc., USA). Sample localities and measurements for each specimen are presented in S1 Table. Measuring methods followed Chen *et al.* [25], Yang *et al.* [53].

Cranial measurements were analyzed by PCA in SPSS v19.0 (SPSS Inc., USA). PCAs were conducted on $log_{10}$ transformed variables on two data sets. Before analysis, the Kaiser–Meyer–Olkin (KMO) test (to check correlations or partial correlations between variables), and Bartlett's sphere test (to determine if the correlation matrix is an identity matrix) were performed.

## Results

### Phylogenetic analysis

We obtained 69 mitochondrial sequences, and 156 nuclear sequences. All *CYTB*, *APOB*. *BRCA1* and *RAG2* files are available in GenBank (GenBank Accession No.: MK962157-MK962237, MN032125-MN032174, MN032175-MN032241, and MN032242-MN032310) (Table 2).

Bayesian reconstruction using *CYTB* revealed eight monophyletic clades of *Episoriculus*, corresponding to *E. macrurus*, *E. soluensis*, *E. leucops*, *E. caudatus*, *E. umbrinus*, *E. sacratus*, and *E.* sp. (Fig 2A). *P. fumidus* clustered with *Chodsigoa* with strong support (PP = 0.95). *E. macrurus* represented a basal lineage of Nectogalini with strong support (PP = 1.00), and *E. soluensis*, *E. leucops*, *E. sacratus* and *E.* sp. formed a separate, strongly supported lineage (PP = 0.95–1.00). *E. umbrinus* and *E. caudatus*, located at the tip of this tree, formed sister clades, that were less-well supported (PP = 0.75).

Bayesian reconstruction using the three concatenated nuclear genes also revealed eight monophyletic clades (Fig 2B), but with a slightly different topological structure to the *CYTB* tree. *P. fumidus* mixed with *Chodsigoa* and *Soriculus*, with weak support. All species of *Episoriculus* formed a monophyletic clade, with *E. macrurus* at its base. Lineages of *E. soluensis*, *E. leucops*, *E. sacratus*, and *E.* sp. were strongly supported (PP = 1.00). *E. umbrinus* and *E. caudatus* formed sister clades at the tree's tip, although support for them was weak (PP = 0.66).

Structures of three individual nuclear gene trees differed from the *CYTB* and three concatenated nuclear gene trees (Fig 3), with some nodes having very low support. *P. fumidus* did not cluster with *Episoriculus*. *E. macrurus*, *E. soluensis*, *E. leucops*, *E.* sp., and *E. sacratus* remained monophyletic with strong support based on *APOB* and *RAG2* (PP = 1.00) genes, and *E. umbrinus* and *E. caudatus* formed a sister group with weak support. *E. sacratus*, *E. umbrinus* and *E. caudatus* were mixed on the tree based on the *BRCA1* gene, with very low support.

### Species delimitation

The topology of *BEAST species' trees differed slightly from those of mitochondrial and nuclear genes (Fig 4). *E. macrurus*, *E. soluensis*, *E. leucops*, *E. sacratus*, and *E.* sp. also had high support in these trees (PP = 1.00). *E. umbrinus* and *E. caudatus* were sister clades, but with weak support in the tree (PP = 0.75). This result did not support recognizing *E. umbrinus* as a distinct species.

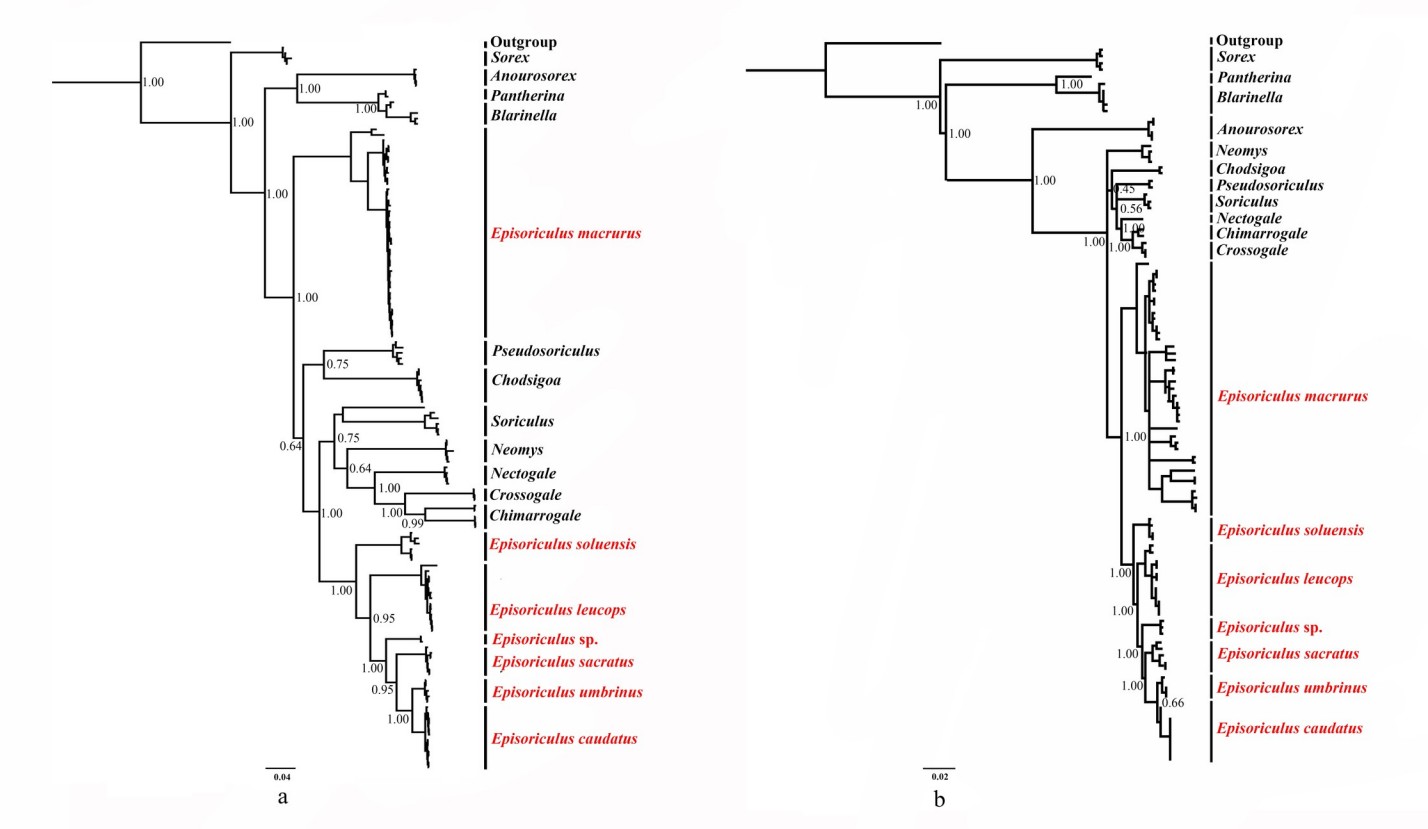

**Fig 2.** Bayesian phylogenetic analyses based on (a) *CYTB*, and (b) three concatenated nuclear genes. Numbers at nodes refer to Bayesian posterior probabilities. Scale bars represent substitutions per site.

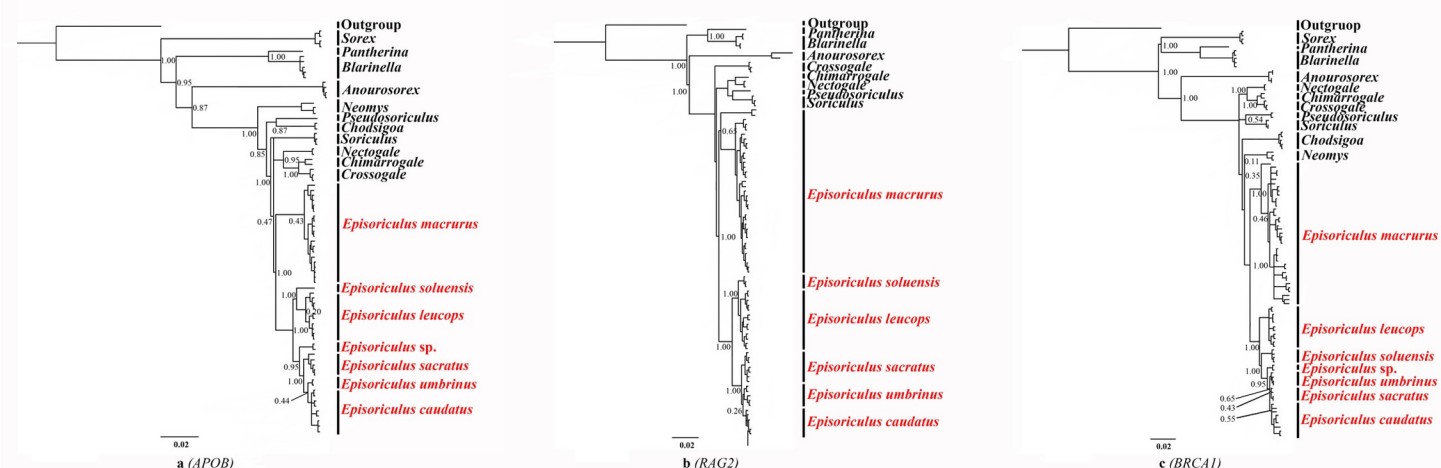

**Fig 3.** Bayesian phylogenetic analyses based on (a) *APOB*, (b) *RAG2*, and (c) *BRCA1* genes. Numbers at nodes refer to Bayesian posterior probabilities. Scale bars represent substitutions per site.

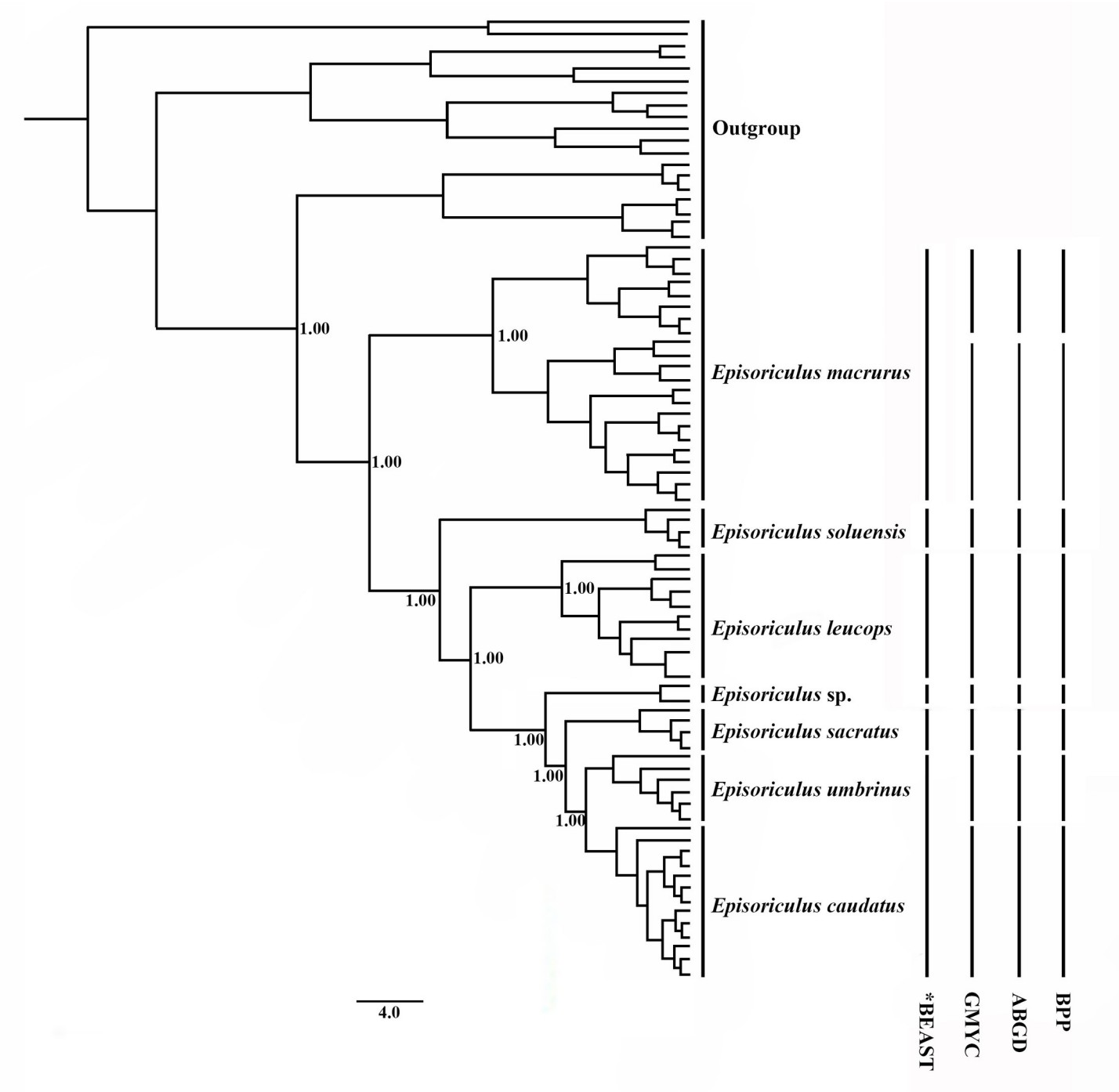

**Fig 4. Results of species delimitation using splits, GMYC, BPP, and species trees reconstructed using the *BEAST model.** Node numbers indicate Bayesian posterior probabilities supporting each clade as two putative species.

GMYC analysis reveals five clades as valid species (Fig 4), of which *E. soluensis*, *E. leucops*, and *E.* sp. are separate, and *E. macrurus* comprised two species, with individuals from Sichuan and Yunnan provinces differing. This analysis suggests that *E. caudatus*, *E. sacratus* and *E. umbrinus* are conspecific.

For the ABGD analysis, the transition/transversion value (3.5) first calculated by Mega 5 was used as the starting parameter, with 0.5, 1.0, 1.5 and 2.0 used as relative gap widths. The 81 samples divided into eight species: *E. caudatus*, *E. macrurus*, *E. sacratus*, *E. soluensis*, *E. leucops*, *E. macrurus* (Sichuan samples), *E. macrurus* (Yunnan samples), and *E.* sp. (Fig 4 and S2 Table).

BPP analysis revealed *E. caudatus* and *E. umbrinus* to be separate species in 12 groups of BPP data based on the combined nuclear gene data set, with support for *E. sacratus* being a valid taxon also being high (Fig 4 and S3 Table).

Kimura-2-parameter (K2p) distances between *Episoriculus* species ranged 0.027–0.160 (Table 5). The average K2p distance between *P. fumidus* and *Episoriculus* species was 0.177. The K2p distance between *E. caudatus* and *E. sacratus* was 0.067, and between *E. caudatus* and *E. umbrinus*, 0.027, and *E. sacratus* and *E. umbrinus*, 0.071.

## Morphology

Morphological data (HBL, TL, HFL, EL, and body weight (BW)) of 56 specimens with intact skulls are presented in Table 6. The TL of *E. macrurus* ranged 71–106mm, while values for congeners ranged 46–82.5 mm. The HFL of *E. macrurus* was 15–16 mm, while for other species it was 12–13 mm. *E. macrurus* had the lowest HBL/TL ratio (0.64), and its TL was ~1.5 times its HBL, while the HBL of congeners was approximately equal to or greater than TL. *E. leucops* had the largest body, and *E. sacratus* differed from *E. caudatus* and *E. umbrinus* in HBL/TL values. Morphological indices are detailed in Table 7, and complete measurement data are provided in S1 Table.

Bartlett's test rejected the null hypothesis ($\chi^2$ = 568.01, P = 0.000), indicating that the data were spherical and variables were somewhat independent of each other. A KMO of 0.813 indicated a strong correlation existed among the various skull data, which was suitable for factor analysis. Two principal components explaining 74.69% of morphological variation were extracted from the analysis. Factor loading values were most positive, indicating that it was mainly related to overall skull size (Table 8). Features with factor loads > 0.8 included PIL, ML, CIL, UTR, and LTR.

Using PC 1 and PC 2 maps (Fig 5), *E. macrurus* plotted in the positive region of PC 2, while other species were mainly in the negative region of PC 2. The larger *E. leucops* plotted in the negative region of PC 2, and *E. soluensis* plotted in the positive region of PC 2. *E. sacratus* was distinguished with *E. umbrinus* and *E. caudatus*, with the latter two species being mixed and not effectively differentiated.

Compared to skulls of other *Episoriculus* species (Fig 6), the braincase of *E. macrurus* is more dome-shaped, the rostrum is shorter; and the upper unicuspids are quadrangular and wider than long (those of other species are similarly sized). Compared to skulls of *E. caudatus*,

**Table 5. The Kimura-2-parameter distances between *Episoriculus* species based on the *CYTB* gene.**

| No. | Species | 1 | 2 | 3 | 4 | 5 | 6 | 7 |
|-----|---------|---|---|---|---|---|---|---|
| 1 | *Episoriculus caudatus* | | | | | | | |
| 2 | *Episoriculus sacratus* | 0.066 | | | | | | |
| 3 | *Episoriculus umbrinus* | 0.027 | 0.071 | | | | | |
| 4 | *Episoriculus* sp. | 0.083 | 0.088 | 0.099 | | | | |
| 5 | *Episoriculus soluensis* | 0.113 | 0.121 | 0.128 | 0.129 | | | |
| 6 | *Episoriculus leucops* | 0.112 | 0.133 | 0.131 | 0.126 | 0.139 | | |
| 7 | *Episoriculus macrurus* | 0.149 | 0.156 | 0.160 | 0.148 | 0.159 | 0.154 | |
| 8 | *Pseudosoriculus fumidus* | 0.181 | 0.184 | 0.192 | 0.177 | 0.163 | 0.189 | 0.154 |

**Table 6. The results of body morphologic measurements.**

| Species | N | BW | HBL | TL | HFL | HBL / TL |
|---|---|---|---|---|---|---|
| *E. caudatus* | 14 | 5.000±0.327 | 56.750±1.858 | 54.750±0.491 | 12.125±0.125 | 1.04 |
| *E. sacratus* | 6 | 5.666±0.211 | 57.134±0.632 | 62.166±2.249 | 13.500±0.671 | 0.92 |
| *E. umbrinus* | 9 | 5.000±0.239 | 61.333±0.816 | 60.222±0.547 | 13.000±0.231 | 1.01 |
| *E. leucops* | 7 | 7.714±0.286 | 62.714±0.056 | 60.143±1.534 | 13.143±0.143 | 1.04 |
| *E. macrurus* | 17 | 5.000±0.413 | 58.158±1.732 | 91.333±1.201 | 15.667±0.333 | 0.64 |
| *E. soluensis* | 4 | 6.000 ±0.210 | 64.333±0.666 | 61.333±0.088 | 12.000±0.531 | 1.04 |

the frontal region of the skull of *E. sacratus* is more arched, and the posterior cusp of its upper incisor is lower than its first unicuspid, whereas the height of the posterior cusp of the upper incisor and first unicuspid of *E. caudatus* are similar.

## Discussion

In the study, we collected a huge number of shrews in China and, via morphological and molecular analysis, concluded that genus *Episoriculus* contains six distinct species and one cryptic species. The majority of *Episoriculus* species have their type origin in India. We further validated the accuracy of our species classification by utilizing sequence collected in India from Ohdachi *et al.* [29]. We did comprehensive studies on some species whose classification is contested.

Thomas [54] described six specimens from Mount Emei as *E. sacratus*, and considered it most likely the local representative of *E. caudatus*, from which they differed in having a much smaller braincase. Allen [55] described the subspecies *E. caudatus umbrinus* from Mucheng, Yunnan, which was most alike *E. sacratus*, but differed from it in its much darker-brown color and in having a uniformly dark rather than bicolor tail. Allen [14] demoted *E. sacratus* to a subspecies of *E. caudatus*—an opinion with which Ellerman and Morrison-Scott [1] agreed. Hoffmann [7] examined many specimens and concluded that there were similarities in skull and cranial size between *E. c. caudatus*, *E. c. sacratus* and *E. c. umbrinus*. Wilson and Reeder [9, 13] recognized these three taxa to be distinct species based on karyotypes and differences in skull size, while Motokawa and Lin [15] considered *E. sacratus* and *E. umbrinus* to be subspecies of *E. caudatus*. Motowaka *et al.* [16] considered the larger *E. caudatus* and smaller *E. sacratus* to be distinct species, and included three subspecies: *E. soluensis*, *E. umbrinus*, and *E. sacratus* as subspecies of *E. caudatus*. Wilson and Mittermeier [3] elevated *E. umbrinus* to full

**Table 7. Morphological measurement data of *Episoriculus* species skulls.**

| Species | N | CIL | IOB | CB | BH | MB | PIL | PPL | UTR | M²-M² | ML | LTR |
|---|---|---|---|---|---|---|---|---|---|---|---|---|
| *E. caudatus* | 14 | 18.327 ±0.057 | 3.311 ±0.044 | 8.769 ±0.374 | 5.907 ±0.068 | 1.391 ±0.018 | 8.104 ±0.059 | 8.067 ±0.039 | 8.1464 ±0.058 | 4.525 ±0.038 | 11.258 ±0.056 | 7.2657 ±0.058 |
| *E. sacratus* | 6 | 18.206 ±0593 | 3.430 ±0.052 | 8.578 ±0.082 | 6.030 ±0.100 | 1.381 ±0.237 | 7.840 ±0.056 | 8.035 ±0.103 | 7.773 ±0.073 | 4.565 ±0.029 | 11.036 ±0.090 | 6.981 ±0.046 |
| *E. umbrinus* | 8 | 18.286 ±0.070 | 3.523 ±0.054 | 8.513 ±0.045 | 5.628 ±0.045 | 1.371 ±0.024 | 8.208 ±0.063 | 8.126 ±0.088 | 8.052 ±0.046 | 4.457 ±0.049 | 11.1789 ±0.077 | 7.3933 ±0.0763 |
| *E. leucops* | 7 | 19.307 ±0.111 | 3.708 ±0.377 | 9.068 ±0.448 | 5.858 ±0.046 | 1.391 ±0.341 | 8.658 ±0.036 | 8.876 ±0.797 | 8.416 ±0.082 | 4.564 ±0.062 | 11.693 ±0.130 | 7.354 ±0.147 |
| *E. macrurus* | 18 | 17.577 ±0.060 | 3.309 ±0.191 | 8.456 ±0.057 | 5.761 ±0.050 | 1.290 ±0.012 | 7.633 ±0.051 | 7.813 ±0.079 | 7.565 ±0.056 | 4.184 ±0.045 | 10.618 ±0.051 | 6.781 ±0.036 |
| *E. soluensis* | 4 | 17.896 ±0.003 | 3.603 ±0.052 | 9.327 ±0.097 | 5.763 ±0.233 | 1.706 ±0.328 | 8.067 ±0.296 | 7.976 ±0.024 | 7.767 ±0.084 | 4.873 ±0.003 | 12.320 ±0.055 | 7.497 ±0.084 |

**Table 8. Character loadings, eigenvalues, and proportion of variance explained by the first two axes (PC 1 and PC 2) of a principal component analysis using the log10-transformed measurements of *Episoriculus*.** The meanings of variable abbreviations are given in the Materials and Methods Section.

| Measurement | Principal component (PC) | |
|---|---|---|
| | 1 | 2 |
| PIL | 0.888 | -0.340 |
| ML | 0.886 | 0.216 |
| CIL | 0.876 | -0.247 |
| UTR | 0.841 | -0.279 |
| LTR | 0.804 | -0.003 |
| CB | 0.750 | 0.290 |
| $M^2$-$M^2$ | 0.733 | 0.502 |
| PPL | 0.704 | -0.409 |
| IOB | 0.682 | -0.258 |
| MB | 0.592 | 0.722 |
| CH | 0.030 | 0.537 |
| Variance explained | 61.022 | 13.664 |

species without reason, while Wei *et al*. [56] regarded it to be a subspecies of *E. caudatus*. Our data support recognizing *E. sacratus* as a valid species, and recognizing *E. umbrinus* to be a subspecies of *E. caudatus*. In morphology, *E. sacratus* can be differentiated from *E. caudatus*, but *E. umbrinus* cannot. Our data do not support the opinion of Motowaka *et al*. [16]. For the two species, we considered that the Jinsha River and Hengduan Mountains shut off communication between *E sacratus* and *E. caudatus*, limiting the species to the edge of the western Sichuan Plateau. And *E. caudatus* is widely distributed in northeastern India, northern Burma, northern Vietnam, and Xizang, Yunnan, Guizhou, China [2, 3, 17, 57].

Gruber [23] described *E. soluensis* and considered it a separate species. Abe [58] reviewed specimens collected in central Nepal and compared them with those from eastern Nepal by Gruber [23], and both suggested that *E. soluensis* was a subspecies of *E. caudatus*, but also possibly synonymous with *E. sacratus*. Hoffmann [7] treated *E. soluensis* as a synonym of *E. caudatus*—an opinion with which Wilson and Reeder [9, 13] and Motokawa and Lin [15] agreed. Ohdachi *et al*. [29] regarded two samples from Nepal to be *E. caudatus soluensis* following Abe [58], and sequenced *CYTB*. Abramov *et al*. [2] then used these two sequences to reconstruct a system tree, and after determining that *E. soluensis* constituted a distinct clade from *E. caudatus*, advocated for them being treated as distinct species. In our tree, four samples from Yadong and Nyalam cluster with the two *E. soluensis* samples of Ohdachi *et al*. [29], and these four specimens are similar in having dark-brown ventral hair, and light-yellowish-brown dorsal hair. The tail length of our four specimens is longer than the head length, the skull parietal bone is relatively protruding, there are four upper single cusp teeth, the posterior cusp teeth of the maxillary incisor are similar in height to the first upper single cusp teeth, and the cusp teeth are light brown. These features are basically consistent with Gruber's [23] original description, and the description of *E. soluensis* of Wilson and Mittermeier [3]. Accordingly, we regard *E. soluensis* to be a distinct species, and report it for the first time from China. And We discovered that this species lives on both sides of the middle Himalayas (Nepal, northeast India, and Shigatse region, China).

While Ellerman and Morrison-Scott [1] regarded *Episoriculus baileyi* to be a subspecies of *E. caudatus*, Abe [58, 59] identified the two to be morphologically distinct and sympatric in Nepal. Hoffmann [7] examined species from Burma and Nepal and considered *E. baileyi* to be

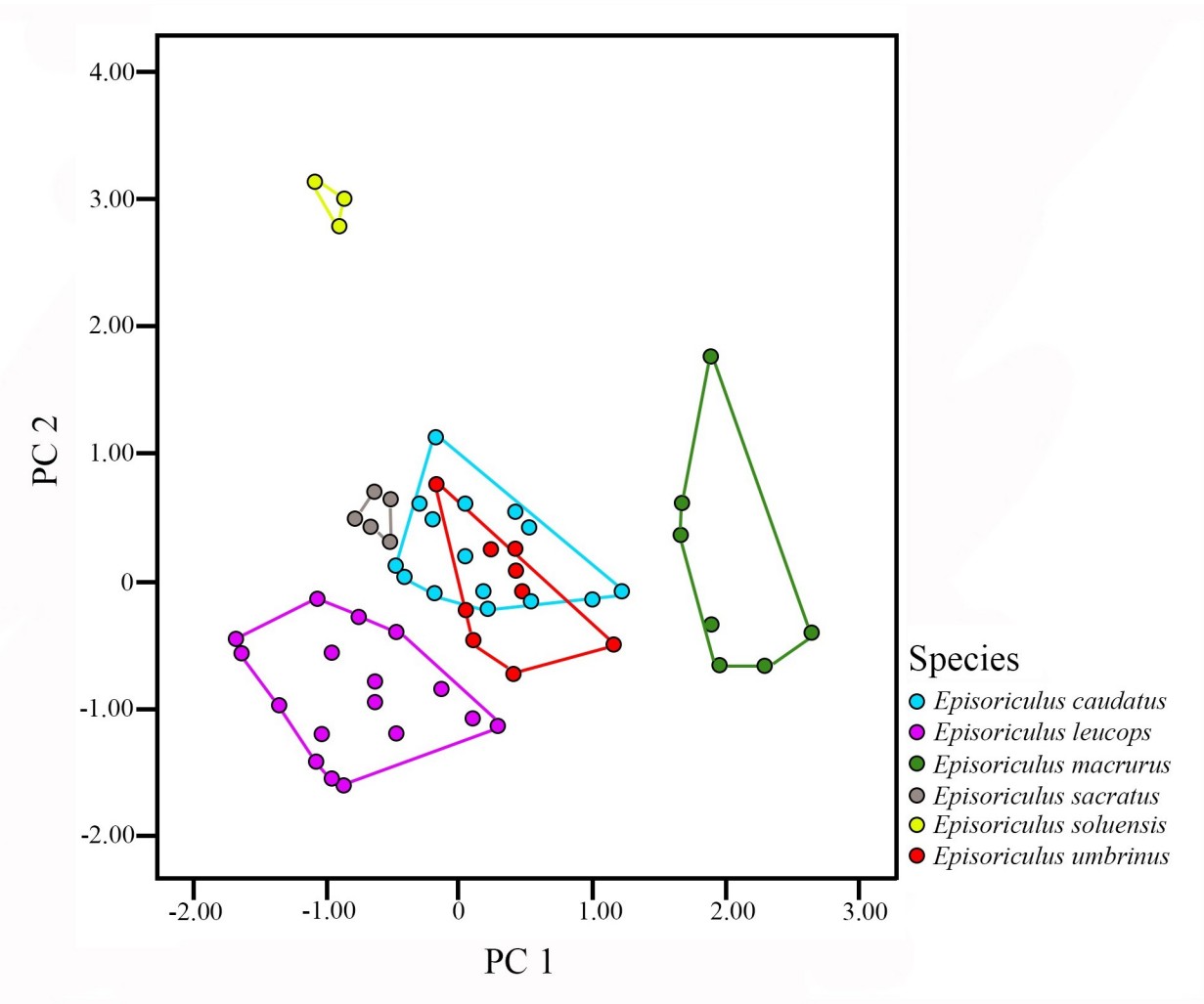

**Fig 5. Results of principal component analysis of *Episoriculus* taxa based on 19 log$_{10}$ transformed craniodental measurements.**

a subspecies of *E. leucops*, an opinion with which Wilson and Reeder [9, 13] agreed. Based on external and cranial morphology, Motokawa and Lin [15] re-evaluated the taxonomic status of *E. baileyi*, and considered it to be a valid species of *Episoriculus*. This species could be distinguished from other *Episoriculus* in the combination of its robust first upper incisor, long rostrum and upper unicuspid row, large tympanic ring, and high ascending ramus of the mandible—an opinion with which Wilson and Mittermeier [3] agreed. Because of a lack of specimens, we cannot investigate the status of *E. baileyi*. We provisionally follow Motokawa and Lin [15], but the taxonomic status of this species requires further investigation.

While Ellerman and Morrison-Scott [1] considered *E. fumidus* to be a subspecies of *E. caudatus*, Jameson and Jones [11] considered it to be a distinct species based on its geographical isolation and morphological divergence—an opinion with which Hoffmann [7], Wilson and Reeder [9, 13], and Motokawa and Lin [15] agreed. Dubey *et al.* [29] inferred that *E. fumidus* (the only representative of the genus in their study) was a sister group of *Chodsigoa* with strong support in the *APOB* gene tree—an opinion with which He *et al.* [17] agreed. Based on the sequence of Dubey *et al.* [29] and He *et al.* [17], Abramov *et al.* [2] regarded *fumidus* do not

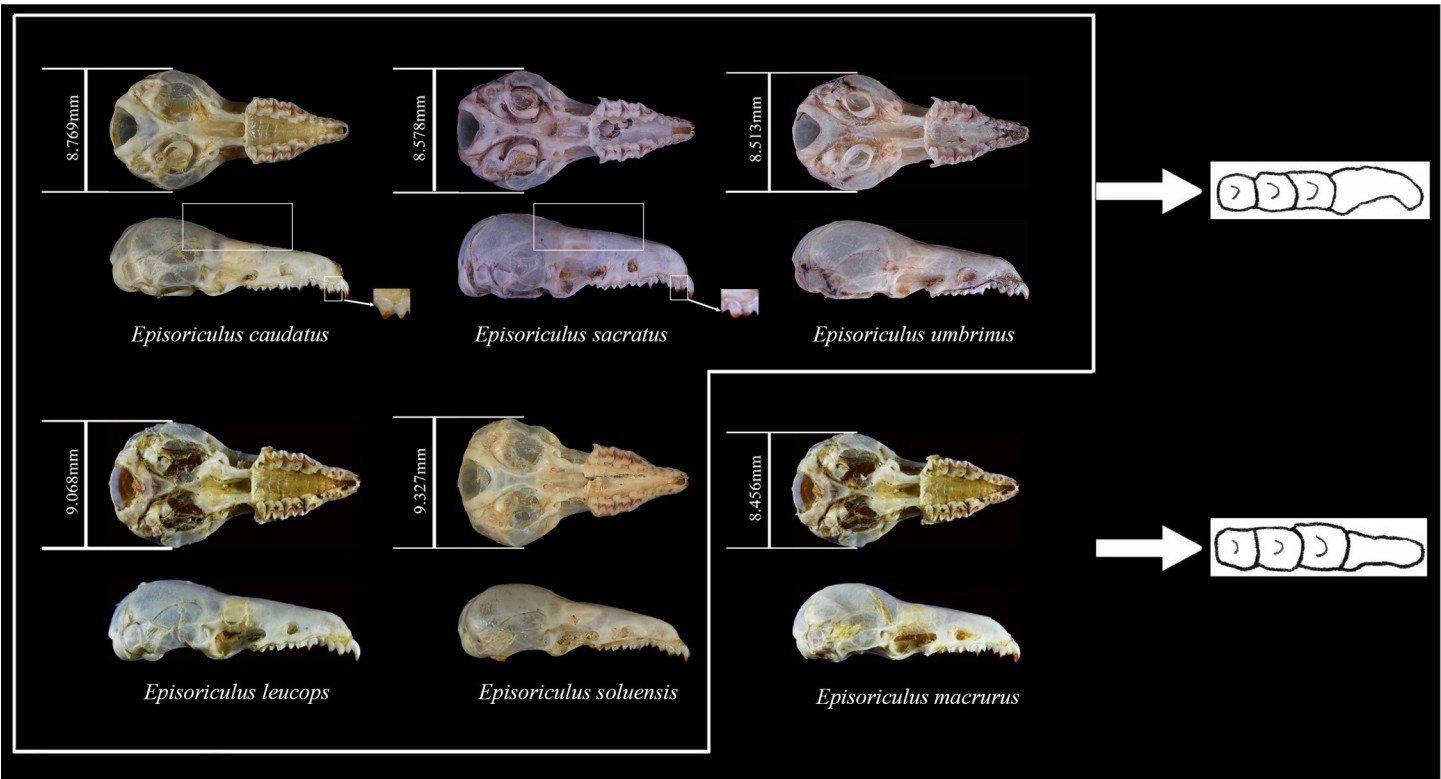

**Fig 6. Comparison of *Episoriculus* skulls.**

belong in *Episoriculus*, and established the genus *Pseudosoriculus* for it—an opinion supported by our analyses.

We identify what appears to be a new cryptic species (*E.* sp.) from low-elevation areas in Motuo County, Xizang, which forms a separate branch in our system tree. However, with only two specimens available, we cannot accurately describe its morphology. Further specimens and molecular data are required to accurately resolve the taxonomic status of this taxon.

Our phylogenic analysis consistently roots *E. macrurus* as an individual lineage. This species, which has large genetic distance from congeners, in phylogenetic trees is usually located in the outermost or most basal part of the genus *Episoriculus*. It has the longest tail in the genus, and differs from congeners in skull and tooth morphology. For these reasons we speculate it retains some of the most primitive traits in genus *Episoriculus* or Nectogalini. However, in phylogenetic trees based on different genes, the position of a varies: in the phylogenetic tree based on mitochondrial gene, *E. macrurus* is at the base of tribe Nectogalini and forms a single monophyletic group, whereas in the nuclear gene-based tree, *E. macrurus* is included in Nectogalini and clustered with other species of *Episoriculus* on the same clade. Similar conflicting phylogenetic signals have been reported in other studies [60]. This phenomenon could be explained from the perspectives of genetic background [61], ancient hybridization [62], incomplete lineage sorting [63], adaptive evolution, and burst Formula speciation [64]. The long tail and developed hind feet of this species lend it a semi-arboreal appearance. This specific niche adaption, which is not shared by other species, may result in huge differences in energy metabolism patterns between *E. macrurus* and other species of Nectogalini. This was reflected in mitochondria, which caused early differentiation in the species tree built using mitochondrial genes, resulting in incomplete lineage sorting. While neither mtDNA nor

nDNA alone resolved phylogenetic relationships in the genus *Episoriculus*, combining data from these two genetic pathways did improve results. Species tree construction in the coalescent framework also produced a consistent topology with high statistical support. Therefore, we deem that a combined approach using mitochondrial and nuclear gene information is more appropriate for resolving phylogenetic relationships in the genus *Episoriculus*.

## Conclusion

Based on molecular and morphological analyses, the genus *Episoriculus* comprises at least six valid species: *E. baileyi*, *E. caudatus*, *E. leucops*, *E. macrurus*, *E. sacratus*, *E. soluensis*, and the potentially undescribed *E*. sp.

## Supporting information

**S1 Table. External and selected cranial measurements of *Episoriculus* species.**
(DOCX)

**S2 Table. Results of ABGD species definition based on *CYTB* gene.**
(DOCX)

**S3 Table. Posterior probabilities supporting three species (*Episoriculus caudatus*, *E. sacratus*, and *E. umbrinus*) as potential species using different algorithms and priors.**
(DOCX)

## Acknowledgments

We are particularly grateful to Robert Murphy for the correct scientific questions. We are also grateful to Yingting Pu and Jiao Qing for their assistance with this study.

## Author Contributions

**Conceptualization:** Yingxun Liu, Shaoying Liu.

**Data curation:** Yingxun Liu, Xuming Wang, Shunde Chen.

**Formal analysis:** Yingxun Liu, Tao Wan.

**Funding acquisition:** Shaoying Liu.

**Investigation:** Yingxun Liu, Xuming Wang, Rui Liao, Shaoying Liu.

**Methodology:** Yingxun Liu, Xuming Wang, Tao Wan, Shunde Chen, Shaoying Liu, Bisong Yue.

**Resources:** Tao Wan, Rui Liao, Shaoying Liu.

**Software:** Yingxun Liu, Xuming Wang, Tao Wan, Shunde Chen.

**Validation:** Shaoying Liu, Bisong Yue.

**Visualization:** Tao Wan.

**Writing – original draft:** Yingxun Liu.

**Writing – review & editing:** Yingxun Liu, Shaoying Liu, Bisong Yue.

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
