## [Decision Letter · Decision Letter 0]

8 Apr 2024

PONE-D-24-05960Integrative phylogenetic analysis of the genus Episoriculus (Mammalia: Eulipotyphla: Soricidae)PLOS ONE

Dear Dr. Liu,

Thank you for submitting your manuscript to PLOS ONE. After careful consideration, we feel that it has merit but does not fully meet PLOS ONE’s publication criteria as it currently stands. Therefore, we invite you to submit a revised version of the manuscript that addresses the points raised during the review process.

Reviewer #2 found several methodological omissions that need to be addressed before considering the manuscript for publication. Such a great step in the taxonomical investigations, that is the revision of the entire genus, must be studiously made by adopting all accessible up-to-date data, without discrimination. The authors are advised to carefully study the reviewers' reports and to carry out the necessary analyses suggested therein. Please be aware of the comments provided by Reviewer #1 in the attached file.

We look forward to receiving your revised manuscript.

Kind regards,

Branislav T. Šiler, Ph.D.

Academic Editor

PLOS ONE

Journal Requirements:

"This research was funded by the National Natural Science Foundation of China (32370496,31970399）"

"This research was funded by the National Natural Science Foundation of China (32370496, 31970399). We are particularly grateful to Robert Murphy for the correct scientific questions. We are also grateful to Yingting Pu and Jiao Qing for their assistance with this study."

"This research was funded by the National Natural Science Foundation of China (32370496,31970399）"

5. Please provide a complete Data Availability Statement in the submission form, ensuring you include all necessary access information or a reason for why you are unable to make your data freely accessible. If your research concerns only data provided within your submission, please write "All data are in the manuscript and/or supporting information files" as your Data Availability Statement.

6. We note that [Figure 1] in your submission contain [map/satellite] images which may be copyrighted. All PLOS content is published under the Creative Commons Attribution License (CC BY 4.0), which means that the manuscript, images, and Supporting Information files will be freely available online, and any third party is permitted to access, download, copy, distribute, and use these materials in any way, even commercially, with proper attribution. For these reasons, we cannot publish previously copyrighted maps or satellite images created using proprietary data, such as Google software (Google Maps, Street View, and Earth). For more information, see our copyright guidelines: http://journals.plos.org/plosone/s/licenses-and-copyright.

Reviewers' comments:

Reviewer's Responses to Questions

**Comments to the Author**

1. Is the manuscript technically sound, and do the data support the conclusions?

Reviewer #1: Yes

Reviewer #2: No

2. Has the statistical analysis been performed appropriately and rigorously? 

Reviewer #1: Yes

Reviewer #2: Yes

3. Have the authors made all data underlying the findings in their manuscript fully available?

Reviewer #1: Yes

Reviewer #2: No

4. Is the manuscript presented in an intelligible fashion and written in standard English?

Reviewer #1: Yes

Reviewer #2: Yes

5. Review Comments to the Author

Reviewer #1: Based on the manuscript presented, it is obvious that the authors have extensively revised their previous submission. The paper now includes more elaborate and sophisticated objective methods for identifying species based on coalescent-based theory.

I am convinced that their combined morphological and molecular analyses are thorough, thoughtful, and appropriate. This manuscript represents a valuable contribution to the taxonomic understanding of this genus in this geographic region. I have included a few relatively minor editorial suggestions, and a couple of suggestions to improve the presentation and clarity of the manuscript.

Reviewer #2: The genus Episoriculus are among of the poorly studied taxa of the tribe Nectogalini due to difficulties of morphological descrimination and wide geographical distribution in Asia — from northern China, southward to northern Vietnam and Myanmar and from Kashmir to Taiwan.

The article claims to be a phylogenetic study and another taxonomic revision of the genus Episoriculus. In the Introduction and Discussion, the authors list in detail the history of the study and taxonomy of Episoriculus, but this does not add to their understanding of the distribution of all the described forms, including those lineages that are common in China.

Main comment

The main serious remark to the entire manuscript is that it is not clear how they determined that their specimens belonged to one or another taxon. P74 lines 62-64 : “specimens collected from within China, 31 were attributed following Wilson & Mittermeier [3], Hoffmann [7], and Smith & Xi”. But it is impossible to determine the species of Episoriculus only using the key. To do this, it is necessary to carefully analyze type localities (the authors do not do this), study types in different museums (the authors do not do this), and, finally, use ALL genetic data on the item available today from the GenBank.

The latter is especially important in this case, but it is interesting that the authors accidentally or intentionally ignore the sequences of the Nepalese samples of Episoriculus from the publication Ohdachi, …, Abe 2006. If they included them in their analysis, it would become clear that what they call Episoriculus sp. from Tibet, according to genetic data – this is E. caudatus according to Ohdachi et al. 2006, Abe 1997, as well as Abramov et al. 2017 (AB175114, AB175115). If so, a form which they call caudatus (from Yunnan mainly), - it is unknown what, close to umbrinus but not true E. caudatus.

I am amazed that while listing the history of the study of for so long and mentioning various authors, including Ohdachi and Abe, they do not use the sequences from Nepal in their work. In general, most species of Episoriculus were described from Nepal and NE India, so having only Chinese data, it is difficult to understand what is what. So, I suggest, this is an unacceptable error in the present work.

Мinor comments

The figures are very sloppy and have poor resolution.

Table 2. Note on the table : “* Indicates that this sequence was downloaded from NCBI.” However, I could not find a asterisks the table!

I don’t think that this manuscript with an incorrect species identification can be published in PlosOne or anywhere else at all.

6. PLOS authors have the option to publish the peer review history of their article (what does this mean?). If published, this will include your full peer review and any attached files.

Reviewer #1: No

Reviewer #2: No

---

## [Author Response · Author response to Decision Letter 0]

16 Jul 2024

Reply on comments of Associate Editor and two anonymous reviewers

Dear Branislav T. Šiler, Ph.D.,

Thank you very much for your comments on our paper. Your comments are very useful for us to improve our manuscript. We consider the reviewer's revision opinions are very essential and meaningful, and we will make faultless adjustments to the paper.

We have carefully revised our paper according to your comments. We hope this revision suitable for publication on PLOS ONE. 

And reply to others as follow.

Reviewers' comments:

Comments to the Author

Reviewer #1: Based on the manuscript presented, it is obvious that the authors have extensively revised their previous submission. The paper now includes more elaborate and sophisticated objective methods for identifying species based on coalescent-based theory.

Viewpoint: I am convinced that their combined morphological and molecular analyses are thorough, thoughtful, and appropriate. This manuscript represents a valuable contribution to the taxonomic understanding of this genus in this geographic region. I have included a few relatively minor editorial suggestions, and a couple of suggestions to improve the presentation and clarity of the manuscript.

Reply: We are very grateful for your recognition on our manuscript. We would like to offer our gratitude to you for your revision ideas. We have altered them one by one, hoping to satisfy your expectations.

Reviewer #2: The genus Episoriculus are among of the poorly studied taxa of the tribe Nectogalini due to difficulties of morphological descrimination and wide geographical distribution in Asia — from northern China, southward to northern Vietnam and Myanmar and from Kashmir to Taiwan.

Question: The article claims to be a phylogenetic study and another taxonomic revision of the genus Episoriculus. In the Introduction and Discussion, the authors list in detail the history of the study and taxonomy of Episoriculus, but this does not add to their understanding of the distribution of all the described forms, including those lineages that are common in China.

Reply: We have contributed to the discussion the distribution of species about Episoriculus and amended the distribution of some species.

Main comment

The main serious remark to the entire manuscript is that it is not clear how they determined that their specimens belonged to one or another taxon. P4 lines 62-64 : “specimens collected from within China, 31 were attributed following Wilson & Mittermeier [3], Hoffmann [7], and Smith & Xi”. But it is impossible to determine the species of Episoriculus only using the key. To do this, it is necessary to carefully analyze type localities (the authors do not do this), study types in different museums (the authors do not do this), and, finally, use ALL genetic data on the item available today from the GenBank.

Reply: Although the morphology of Eulipotyphla species is usually indistinguishable, the morphological differences of each species in genus Episoriculus are very obvious and easy to distinguish, and we could identify species in Episoriculus by combining the morphological identification characteristics provided by the species' original literature and other important literature. Simultaneously, for all collected species, we used the CYTB gene to ensure species accuracy. This process will be further explained in this article.

Question: The latter is especially important in this case, but it is interesting that the authors accidentally or intentionally ignore the sequences of the Nepalese samples of Episoriculus from the publication Ohdachi, …, Abe 2006. If they included them in their analysis, it would become clear that what they call Episoriculus sp. from Tibet, according to genetic data – this is E. caudatus according to Ohdachi et al. 2006, Abe 1997, as well as Abramov et al. 2017 (AB175114, AB175115). If so, a form which they call caudatus (from Yunnan mainly), - it is unknown what, close to umbrinus but not true E. caudatus.

Reply: We supplemented the sequence of Ohdachi et al. (2006) and recalculated the phylogenetic relationships of the genus Episoriculus. We determined that E. umbrinus is not a separate species, but as a subspecies of E. caudatus, and E. soluensis is a separate species. 

Question: I am amazed that while listing the history of the study of for so long and mentioning various authors, including Ohdachi and Abe, they do not use the sequences from Nepal in their work. In general, most species of Episoriculus were described from Nepal and NE India, so having only Chinese data, it is difficult to understand what is what. So, I suggest, this is an unacceptable error in the present work.

Reply: We think it's critical that you bring up this error. We employed only a few sequences of Ohdachi et al. (2006) in our earlier study. Sequence samples from this genus were gathered in Nepal, providing more accurate evidence for species classification when used in sequence analyses. In molecular analysis of the manuscript, we supplemented the relevant sequences, reconstructed the phylogenetic tree, and species delimitation.

Minor comments

The figures are very sloppy and have poor resolution.

Reply: We changed all of the photographs with poor resolution issues to try to portray the information offered by the images in high definition.

Table 2. Note on the table : “* Indicates that this sequence was downloaded from NCBI.” However, I could not find a asterisks the table!

Reply: I'm so sorry. It was our mistake. We have corrected it in the manuscript

I don’t think that this manuscript with an incorrect species identification can be published in PlosOne or anywhere else at all.

Reply: We respeciated all specimens obtained and utilized in this paper, and corrected certain samples that had inaccuracies.

Comments in the manuscript:

Line 17 Change “relegated to” to “assigned”

Reply: We agree with you and have revised it in the manuscript.

Line 37 add “sequences” 

Reply: We agree with you and have added it in the manuscript.

Line 46 Change “the ” to “A”, and “species”to “taxon”

Reply: We agree with you and have revised it in the manuscript.

Line 80 I do not see an asterisk anywhere in the Table above.

Reply: We sorted the sequences downloaded from NCBI into Table 4. This comment has now been removed.

Line 120 Change “heterozygo” to “heterozygous”.

Reply: We agree with you and have revised it in the manuscript.

Line 135 Provide a reference regarding the use of “adductive theory” in phylogenetic. Do you simply mean the most parsimonious explanation.

Reply: We supplement the literature and simple explanation on coalescent theory.

Line 136 “Phased”? Meaning?

Reply: Since the *BEAST model requires that every gene segment in each sample be complete, we didn't use all samples. Our initial expression used partial sequences for analysis, but the wording was incorrect and has since been rectified.

Line 150 Commented [A4]: “Progress”?

Reply: We want to express the notion of application here, the terminology is inappropriate, thus we will edit and omit the word 'progress'.

Line 167 Commented [A5]: Do you mean sexual dimorphism

Reply: Yes. We think the word you used is more accurate, so replace it here.

Line 168 Commented [DS6]: A useful reference to cite here is Zidarova, S. (2015). Is there sexual size dimorphism in shrews? A case study of six European species of the family Soricidae. Acta Zoologica Bulgarica, 67(1), 19-34. This paper concludes that there is a low degree of sexual size dimorphism among members of the family Soricidae.

Reply: Based on the research provided by the reviewers, we confirmed that red-toothed shrews, including genus Episoriculus, did not display distinct sexual differences in size. We will cite this paper as a reference.

Line 177 Commented [A7]: I’m not familiar with this variable,“condylox”?

Reply: Condylox incisive length is profile length. For ease of understanding, we have changed the name of this measurement data.

Line 288 Commented [DS8]: I suggest starting off the discussion with what your general conclusion will be. A statement something like “Based on a combination of molecular and morphological analyses, including multiple coalescent-based approaches to identifying species using DNA sequence data, we conclude that the genus Episoriculus comprises at least six valid species: E. baileyi, E. caudates, E. leucops, E. macrurus, E. sacratus, E. soluensis, and the potentially undescribed E. sp. That said, there are some inconsistencies among the various data sets analyzed, and these will be discussed below

Reply: We agree with your viewpoint and have modified the logical sequence of the discussion parts.

Line 289 Commented [A9]: Briefly discuss each of these in turn. For example, your results describe differences in genetic background. How does this explain the differences?

Reply: In this paragraph, we have simply provided a few plausible explanations for this event. We confirm that this is due to incomplete lineage sorting in the study, hence we will just elaborate on this one factor.

Line 295 Commented [DS10]: Anadromous is not the correct word choice here. Perhaps “coalescent framework” would be better?

Reply: We agree with you and have replaced the word.

---

## [Editor Report · Decision Letter 1]

22 Jul 2024

PONE-D-24-05960R1Integrative phylogenetic analysis of the genus Episoriculus (Mammalia: Eulipotyphla: Soricidae)PLOS ONE

Dear Dr. Liu,

Thank you for submitting your manuscript to PLOS ONE. After careful consideration, we feel that it has merit but does not fully meet PLOS ONE’s publication criteria as it currently stands. Therefore, we invite you to submit a revised version of the manuscript that addresses the points raised during the review process.

**I find the reviewers' concerns fairly addressed, as the phylogenetic relationships have been recalculated and the text amended according to their suggestions. In their revision, the authors have left brackets in L65 empty: **L65: "After euthanasia with eugenol, following the ASM guidelines ()".**Please fill the brackets with the citation and provide the reference in the final list.**

We look forward to receiving your revised manuscript.

Kind regards,

Branislav T. Šiler, Ph.D.

Academic Editor

PLOS ONE
---

## [Author Response · Author response to Decision Letter 1]

7 Nov 2024

Dear Branislav T. Šiler, Ph.D.,

We are grateful for your email and suggestions for our manuscript "Integrative phylogenetic analysis of the genus Episoriculus (Mammalia: Eulipotyphla: Soricidae) " (PONE-D-24-05960R1). Following your comments, we have made the corrections accordingly in the manuscript.

We have re-uploaded figure 1 in compliance with copyright regulations. We uploaded the CC BY 4.0 license file, in which we obtained the permission and signature of the image owner. We have made some revisions to the references. If the permission is not approved, we considered that it is feasible to delete Figure 1, which will not reduce the quality of the manuscript, avoid copyright disputes, and speed up the review progress. And we confirm that laboratory protocol is not suitable for our study, because there is uniqueness in sampling, and other research groups cannot collect the same samples. But we also submitted a laboratory protocol in other section according to the format for reference.

We hope that we can meet the requirements of modification this time.

Best regards,

Liu Yingxun

ASSOCIATE EDITOR COMMENTS:

1. I find the reviewers' concerns fairly addressed, as the phylogenetic relationships have been recalculated and the text amended according to their suggestions. In their revision, the authors have left brackets in L65 empty: L65: "After euthanasia with eugenol, following the ASM guidelines ()".

Please fill the brackets with the citation and provide the reference in the final list.

Response: We have filled the brackets with the citation and added the reference in the final list.

2. We note that [Figure 1] in your submission contain [map/satellite] images which may be copyrighted. All PLOS content is published under the Creative Commons Attribution License (CC BY 4.0), which means that the manuscript, images, and Supporting Information files will be freely available online, and any third party is permitted to access, download, copy, distribute, and use these materials in any way, even commercially, with proper attribution. For these reasons, we cannot publish previously copyrighted maps or satellite images created using proprietary data, such as Google software (Google Maps, Street View, and Earth). For more information, see our copyright guidelines: http://journals.plos.org/plosone/s/licenses-and-copyright.

Response: We have re-uploaded figure 1 in compliance with copyright regulations. 

We also uploaded the CC BY 4.0 license file, in which we obtained the permission and signature of the image owner.

Response: We have made some revisions to the references.

JOURNAL REQUIREMENTS:

Response: We have reviewed the references, and determined that no references have been withdrawn or unpublished.

2. While revising your submission, please upload your figure files to the Preflight Analysis and Conversion Engine (PACE) digital diagnostic tool, https://pacev2.apexcovantage.com/. PACE helps ensure that figures meet PLOS requirements. To use PACE, you must first register as a user. Registration is free. Then, login and navigate to the UPLOAD tab, where you will find detailed instructions on how to use the tool. If you encounter any issues or have any questions when using PACE, please email PLOS at <a href="mailto:figures@plos.org">figures@plos.org. Please note that Supporting Information files do not need this step.

Response: We have uploaded and archived the six figures to PACE and modified it as required.

---

## [Editor Report · Decision Letter 2]

16 Dec 2024

Integrative phylogenetic analysis of the genus Episoriculus (Mammalia: Eulipotyphla: Soricidae)

PONE-D-24-05960R2

Dear Dr. Liu,

We’re pleased to inform you that your manuscript has been judged scientifically suitable for publication and will be formally accepted for publication once it meets all outstanding technical requirements.

Kind regards,

Ishtiyaq Ahmad, Ph.D

Academic Editor

PLOS ONE

---

## [Editor Report · Acceptance letter]

8 Jan 2025

PONE-D-24-05960R2 

PLOS ONE

Dear Dr. Liu, 

I'm pleased to inform you that your manuscript has been deemed suitable for publication in PLOS ONE. Congratulations! Your manuscript is now being handed over to our production team.

Kind regards, 

on behalf of

Dr. Ishtiyaq Ahmad 

Academic Editor

PLOS ONE